# Models of the Gene Must Inform Data-Mining Strategies in Genomics

**DOI:** 10.3390/e22090942

**Published:** 2020-08-27

**Authors:** Łukasz Huminiecki

**Affiliations:** Department of Molecular Biology, Institute of Genetics and Animal Biotechnology, Polish Academy of Sciences, 00-901 Warsaw, Poland; l.huminiecki@igbzpan.pl

**Keywords:** gene concept, scientific method, experimentalism, reductionism, anti-reductionism, data-mining

## Abstract

The gene is a fundamental concept of genetics, which emerged with the Mendelian paradigm of heredity at the beginning of the 20th century. However, the concept has since diversified. Somewhat different narratives and models of the gene developed in several sub-disciplines of genetics, that is in classical genetics, population genetics, molecular genetics, genomics, and, recently, also, in systems genetics. Here, I ask how the diversity of the concept impacts data-integration and data-mining strategies for bioinformatics, genomics, statistical genetics, and data science. I also consider theoretical background of the concept of the gene in the ideas of empiricism and experimentalism, as well as reductionist and anti-reductionist narratives on the concept. Finally, a few strategies of analysis from published examples of data-mining projects are discussed. Moreover, the examples are re-interpreted in the light of the theoretical material. I argue that the choice of an optimal level of abstraction for the gene is vital for a successful genome analysis.

## 1. Introduction

The gene is one of the most fundamental concepts in genetics (It is as important to biology, as the atom is to physics or the molecule to chemistry.). The concept was born with the Mendelian paradigm of heredity, and fundamentally influenced genetics over 150 years [1]. However, the concept also diversified in the course of its long evolution, giving rise to rather separate traditions in several sub-disciplines of genetics [2]. In effect, somewhat different narratives about the gene (and models of the gene) developed in classical genetics, population genetics, molecular genetics, genomics, statistical genetics, and, recently also, in systems genetics.

The fundamental goal of this paper is to summarize the intellectual history of the gene, asking how the diversity of the concept impacts on data integration and data-mining strategies in genomics. I hope to show that many practical decisions of a statistician, a bioinformatician, or a computational biologist reflect key theoretical controversies that permeated the field of genetics for over a century (Many practical tasks must be informed by the theory.). When integrating data or designing databases or writing software, an analyst must make smart decisions about the architecture of their system. For example, a genomics database may focus on low-level concepts such as exons, or individual transcripts. Alternatively, a database may focus on higher-level concepts such as gene families, pathways, or networks.

There are two parts to this text. In the first part, which is theoretical, I discuss the background of the gene concept in history and ideas. The theoretical part will set the ground for more practical considerations. In the second part, several practical examples of genome analyses from my own work will be given (The examples will be re-interpreted in the light of the theoretical concepts introduced in the first part.).

### 1.1. Ideas and Concepts—The Development of the Mendelian Paradigm of Heredity

The story of the gene started with Gregor Mendel, a Moravian monk, scientist, and later abbot, who hypothesized an innovative model of particulate heredity. In Mendel’s model, independent intracellular elements determine differentiation of cells into visible traits. Importantly, Mendel’s particles of heredity do not mix with each other, instead, they persist over generations (However, new varieties can evolve through mutations).

In many ways, Mendel was a prophet of genetics in the 19th century. In the first place, he hypothesized a particulate paradigm of heredity (The paradigm dominated genetics in the 20th century, complementing Darwin’s theory of evolution, and developing further with molecular biology and genomics.). Moreover Mendel was an early experimentalist, relying on prospective experimental procedures to test his hypothesis [1]. This was a rather modern approach in the century still dominated by an observational tradition. Interestingly, Mendel’s laboratory was exceedingly simple—it was just the vegetable garden of his monastery. At the same time, Mendel excelled in the scientific method (including logico-mathematical analysis). This enabled him to design just the right simple experiment and to interpret its results insightfully.

At first, however, the Mendelian model of the gene was just a black box that could not be looked into to discern the molecular mechanism. The black box was perfectly non-transparent—nothing was formally postulated about parts of the model, or how these parts interacted. The model specified that the gene did certain things for the cell, but not how. In effect, the gene model was a mere instrument of analysis. It worked best as an intermediate variable in the context of breeding or medical genetics.

Mechanistic details about the gene were only provided when molecular biology revolutionized classical genetics. As a result, genetical phenomena became more akin physical phenomena. This precipitated an avalanche of discoveries. After a physical form was given to Mendel’s abstract gene concept, biochemical details of molecular processes such as transcription and translation, as well as of their regulation, were being revealed [3].

In addition to molecular biology, ingenious experimental models were also crucial. This commonly meant a careful choice and laboratory upkeep of a particular animal species, a plant species, or a microorganism that facilitated a certain class of genetical experiments. For example, Max Delbrück, who was a physicist with interest in genetics [4], developed simple experimental models in viruses and bacteria (Note that the term *model* in this case has no mathematical or conceptual meaning, but instead signifies a simple biological system established for the purposes of experimental study. That the above was achieved by a physicist is not surprising, as there had been a long tradition of working with simple but enlightening models in physics.).

### 1.2. Genomics Technology and the Gene

It might be generally argued that progress in genetics was facilitated by enabling experimental technologies. This was the case for protocols of molecular biology as mentioned above, however, the same may also be argued for genomics. Like molecular biology before, genomics also challenged many orthodox views of the Mendelian gene [5,6]. For example, genomics revealed that genes are not always independent of each other, as they may overlap in complex loci [7].

Presently, assays based on next generation sequencing, followed by bioinformatics, and high-dimensional statistical analysis are drivers for progress in the emergent discipline of systems genetics. At the same time, the high-throughput assays are not the panacea. Next generation sequencing may promote a technology-focused culture, with most attention invested in data generation, data post-processing and storage, as well as descriptive analysis of high-throughput datasets. With the focus on the high-throughput technology, formulation and critique of novel biological concepts receives less attention. In effect, high-throughput datasets are increasingly easy to generate but difficult to interpret.

### 1.3. Empiricism

Let me now introduce several methodological strategies grouped under the umbrella of the method of science. This is crucial to deeper intellectual understanding of the story of the gene. The strategies are as follows: empiricism, experimentalism, reductionism, anti-reductionism, statistical data analysis, and scientific model building.

Empiricism is a philosophical theory that was instrumental to the emergence of the scientific method and the success of science. According to empiricism, all knowledge, including scientific knowledge, can only have solid grounding in the generalizations of practical experiences derived from the senses. Empiricism is skeptical about the value of non-scientific prior knowledge such as systems of beliefs associated with religions, or even views and theories of philosophical schools.

In biology, the empirical tradition of observing and collecting samples from the natural world goes all the way back to the first biological studies of Aristotle [8], which focused on animals (their parts, movements, generation, and development). Aristotle was an ancient but pragmatic philosopher in the Academy of Classical Athens (4th century BC). He was one of the most able students of idealistic Plato. However, Aristotle was much more empirically minded than Plato. Plato himself was skeptical about the value of sensory observations favoring the sources of knowledge that were alternative to empiricism but still based on rational thinking. Plato favored either logical deductions from abstract theories, or instinctive reasoning known presently as intuition (Note that intuition is built from past experiences using unconscious functionalities of the brain that are probably analogous to machine learning.). On balance, Aristotelianism proved more important for the development of empiricism in genetics than Platonism. However, Aristotelian logical doctrines eventually became synonymous with medieval intellectual stagnation, scholasticism, and a lack of scientific creativity.

### 1.4. The Baconian Method

Francis Bacon, a 17th century English philosopher and statesman, formulated an ideological and political manifesto calling for the abandonment of Aristotelianism. His proposal was contained in the treaty *Novum Organum* [9]. In particular, Bacon called for the rejection of futile deductions, unproductive syllogisms, and scholasticism of late Aristotelianism. Instead, Bacon affirmed systematic empiricism on a grand scale, supported by the state politically and financially (Bacon made many insights into the political organization of the scientific system. Indeed, Bacon’s motivation for promoting empiricism politically might have been to lay the grounds for the economic development of the English state.). Interestingly, Bacon is not known for making grand original scientific discoveries like Copernicus, Galileo, or Newton. Instead, his main legacy lies in the vision of science as a systematic body of empirical knowledge and a political program.

Methodologically, Bacon underlined that only unbiased empirical observations can yield certain knowledge. Facts might be gathered selectively to support prior ideas if empiricism is not performed according to a strict plan. Moreover, generalizations, proceeding by the method of induction (named so in contrast with Aristotelian deduction), must not be made beyond what the facts truly demonstrate. As the reader may easily guess, the repeating cycles of: (1) systematic empirical observations, and (2) inductive generalizations—taken together—form the basis of the Baconian method. However, the transition to the method of Bacon was a slow and gradual process. This was probably because, the Baconian enterprise required the professionalization of science and the development of state funding, which became possible only in the 20th century (The observational tradition, rather than experimentalism, continued to dominate the study of nature in the 19th century. Empirical evidence was given, but it tended to be unsystematic and anecdotal, chosen arbitrarily, frequently in the form of samples casually collected by gentlemanly hobbyists. There was little or statistical analysis of data. Moreover, 19th century scientific theories tended to be sweeping and somewhat over-ambitious in relation to limited empirical support. Examples include Charles Lyell’s and Charles Darwin’s overarching treaties, as both the authors were upper-class men whose inspirations were mostly travels, conversations, reading, or correspondence.).

### 1.5. The Baconian Method and Integrative Genomics: On the Importance of Identifying and Avoiding Bias

In our age, genomics is a new example of the need for the Baconian principle of unbiased empiricism. Note that bias could be introduced on many levels. One type of bias may result from temporarily-, spatially-, or taxonomically-limited sampling of the genome space. For example, scientists sequencing extant genomes are only probing the biosphere as it exists at present. Their generalizations may not be applicable to conditions on Earth in the near past, e.g., during the Last Glacial Period, when different selection forces acted on the population. Even more so, their generalizations are unlikely to be applicable to the distant past when the composition of the atmosphere (and, therefore, of the biosphere) was substantially different. For example, recall that oxygen levels in the atmosphere where at the maximum of around 30% about 280 million years ago, and practically zero before the Great Oxygenation Event approximately 2.4 billion years ago. Even if one focuses exclusively on the present conditions, one should appreciate that certain spatial regions of the biosphere are under-sampled. For example, the deepest parts of the ocean and its floor, or deeper parts of Earth’s crust are still physically difficult to access. Finally, care should be taken not over-generalize beyond the taxa from which genomic data were obtained, or for which it is parsimonious to extend the findings based on phylogenetic relationships.

Other sources of bias may result from the limitations of the experimental technologies of genomics. For example, the technological platform of microarrays, used for expression profiling, can only measure expression of genes for which probes were pre-selected by the manufacturer of the platform. Until relatively recently, this meant a set of protein-coding genes of a given species. (At present, however, that there are also commercial microarrays targeting *micro*RNAs). Moreover, as the features of the microarray chip are arbitrarily pre-selected, there could be a bias towards well-studied and highly transcriptionally active genes (Such *active* genes are also called highly expressed.). It is logical that genes that are weakly, or temporarily expressed, or expressed in a very tissue-specific manner were less likely to be discovered using conventional techniques of molecular biology (and pre-selected for the inclusion on the chip). Additionally, most microarray chips are not designed to discern between alternative transcripts (In practice, this usually means that the probe or probes on the chip are designed to target the longest transcript from a reference collection, such as RefSeq [10].).

### 1.6. Experimentalism and the Laboratory

In the 20th century, there was a social and methodological change in empiricism. Most scientists became professionals, working in academic or governmental laboratories. Research became routinely funded through dedicated grant agencies, which were becoming independent of the government. Women were increasingly involved in the profession. Moreover, casual observations were increasingly replaced by systematic scientific experiments. Unlike observations of naturalists, scientific experiments were planned in advance, being designed to test an explicitly stated hypothesis. The setting was controlled, either in field conditions, or in carefully managed laboratories. Moreover, confounding variables were controlled for by assigning test subjects randomly to either experimental groups or the control. There also emerged technical protocols for biochemistry and molecular biology, highly specialized laboratory chemicals and biological reagents, as well as advanced laboratory instrumentation (for example, an electron microscope). Thus, natural philosophy was being replaced by experimental natural science.

With the rise of experimental biology, a new type of heroic scientific figure emerged: an experimentalist. One of the most outstanding experimentalists in genetics was Thomas Hunt Morgan. He was originally an embryologist rather than a geneticist. However, Morgan moved later on in his career to test experimentally several components of the chromosomal theory of heredity. Morgan’s greatest strength was in the ability to set up sophisticated genetical experiments in well-chosen and expertly run animal models. For example, Morgan worked on genetical problems in several experimental animal models that he mastered in his laboratory, especially in the fruit fly [11]. A further strength of these experiments lied in the fact that they were quantitative; for example, Morgan not only demonstrated inheritance linked to sex chromosomes, but also constructed first quantitative chromosomal maps (Morgan’s student used frequencies of crossing-over between six sex-linked loci on the X chromosome of the fly as the proxy measure of chromosomal distances [12].).

### 1.7. The Theory of Experimentalism

The work of practical experimentalists such as Morgan was complemented by the development of the theory of empirical knowledge. A set of increasingly well-understood practices, known presently as the scientific method, was being developed, studied, and codified. The scientific method integrated a range of disparate tasks, including: (1) constructing scientific instruments, (2) formulating hypotheses and designing experiments, (3) making observations and recording results, (4) statistical data analysis and interpretation, (5) modeling and formulating generalizations or new scientific laws, and (6) developing theories about how science works.

Note that early theoretical insights into experimentalism tended to be made by applied statisticians. For example, an English statistician, Sir Ronald Aylmer Fisher pioneered many practical methods for the analysis of experimental datasets, and made advances in the theory of the design of experiments, randomization, and the optimal sample size [13].

In parallel, a group of German-speaking philosophers known presently as logical empiricists [14] formed an academic and social movement devoted to promoting science as a social cause and a set of methodological doctrines (The members of these social groups met in 1920s and 1930s in the so-called Vienna and Berlin circles. Later, many of them emigrated to Anglo-Saxon countries, where they were active in 1940s.). Logical empiricists promoted the idea that experiments must be intertwined with logical analyses [14]. In other words, there was a broad understanding among logical empiricists that experimentalism was no mindless collecting of facts. According to logical empiricists, a scientist advances new hypotheses through logical analyses of old theories and data. The best hypotheses are then prioritized for experimental testing. Logic is then employed again to develop insightful interpretations of results.

Of course, the theory of experimentalism has continued to develop since the peak of the activity of logical empiricists. Philosophers of the following generations argued critically that logical empiricism presented a sort of sterile, overly idealized vision of the scientific method. In particular, Thomas Kuhn argued that social and historical factors must also be taken into account. He argued that science effectively developed in a series of socially-conditioned paradigm shifts [15], in which dominant personalities, fashions, or the socio-economical context could be just as important as scientific methods and facts.

At the end of this section, I would like to argue that the need for interplay between empiricism and logical analysis is well illustrated in the story of the gene. For example, Mendel’s experimental results would have probably been completely overlooked if there were not accompanied step-by-step by insightful logical analyses. First, Mendel proposed, inspired by intuition, that discrete particles were the bearers of hereditary information. In the second step, the experiments on hybrids of varieties of *Pisum sativum* were designed. These two initial steps were logico-analytical. In the next step, the model was verified empirically. In the final step, the implications of confirmatory results were logically considered by Mendel, leading to the crystallization of his paradigm of heredity.

Moreover, Mendel’s theory is also an example of a socially-conditioned scientific paradigm shift of the type postulated by Thomas Kuhn. The Mendelian theory was a radical intellectual revolution replacing pre-scientific views on heredity dominant until the 19th century, in particular the blending theory of inheritance [16]. Gregor Mendel was uniquely positioned to lead the revolution having had education in physics and philosophy, and social background in agriculture. The monastery supported his education and experiments. Note that Mendel’s mathematical skills were probably developed when studying physics. In contrast, the strengths of Charles Darwin and his fellow English naturalists were not strong in mathematical skills and invested their energies in travels and sample collecting.

Presently, some theoreticians like to argue that a new paradigm shift in genetics is on the horizon. According to this view, the Mendelian paradigm may need to be updated to accommodate new genome-wide evidence for adaptive mutations, as well as data generated in the field of trans-generational epigenetic inheritance [16]. There can be little doubt that data-mining of genomic datasets will play a key role in this process.

### 1.8. Reductionism

In the history of 20th century genetics, molecular biology played the role of a reducing theory with respect to classical genetics [17,18]. In the case of molecular biology, reductionism [2] means physicalism (Note that there is a related idea in reductionism that breaking a system down into smaller and smaller modules will enable a biologist to understand the large system in fullness. For this to work, the system under consideration needs to be modular, i.e., composed of independent and self-contained units.). Physicalism is a claim and a research program suggesting that biological phenomena can be described and explained first as chemical facts, and in a further reductive step, using the laws of physics, also as physical facts.

Moreover, physicalism was interpreted as the basis for scientific and social progress by the movement for the Unity of Science (this was a group within logical empiricism led by socially-active Otto Neurath). In this radical perspective, empiricism marches towards a comprehensive scientific conception of the world, where all experimental knowledge is unified by low-level chemical and physical principles. If this were true, the molecular interpretation of the gene would need to have the power to explain everything that the classical gene could, but do this with more detail and correctness. However, is this really the case? The difficulty lies in the fact that radical reductionism is a research program with a fundamentalist agenda. The agenda extends beyond a pragmatic need for providing useful molecular facts illuminating old biological problems. Radical proponents of reductionism would like for all biological explanations and research programs to progressively follow the reductionist pattern set by biochemistry and molecular biology. Ultimately, radical reductionists desire all scientific theories to be based exclusively on physicalism. In the opinion of most theoreticians and philosophers, this is neither possible nor desirable in genetics. The following section explains why this is the case.

### 1.9. Anti-Reductionism

Anti-reductionists, who include most philosophers of biology, oppose the ideology of radical reductionism. In particular, anti-reductionists view physicalist explanations as mostly unnecessary, frequently unrealistic, and sometimes even dangerously misleading. Anti-reductionists claim that a scientist dogmatically following the physicalist principle would ultimately develop a kind of conceptual myopia. That is to say, a radical follower of the ideology of reductionism would obsess about details, but fail to see the big picture. To use a popular idiomatic expression, they could not see the forest for the trees. In fact, a dogmatic reductionist might be tempted to claim that there is no such thing as a forest, that every tree must be considered in isolation and on its own. 

It is, therefore, clear that by logical extension the ideology of radical reductionism would lead to a kind of conceptual eliminativism toxic to scientific discourse. For example, an eliminativist might claim that the concept of the gene should be abandoned altogether, in favor of the exclusive focus on the function of individual DNA base pairs. Metaphorically speaking, the elimination of all higher-level concepts would be akin to a massive cerebral infarction. Genetics based on such principles would be merely a mindless collecting of facts. In genetics, there are many examples of useful high-level concepts that are too far removed from the physics of a single atom or the chemistry of a single molecule for reductionism to be a useful approach. Examples include quantitative traits, multi-genic diseases, enzymatic pathways, signaling networks, pleiotropy, and complex loci.

Anti-reductionists can forcefully claim that the high-level concepts must not only be retained, but become the focus of analysis in order to interpret irreducible complexity. Presently, anti-reductionism is becoming equal in importance in genetics to the reductionism of molecular biology. This is leading to the development of a new sub-discipline—systems genetics. Systems geneticists use high-level concepts such as signaling pathways and networks to interpret genome-wide association studies.

### 1.10. Statistical Data Analysis

Statistical tools are essential for the organization, display, analysis, and interpretation of empirical data. It is, of course, beyond the scope of this paper to discuss the development of statistics. Historical material is widely available [19]. I will limit myself to noting that early tools of the biometric school, developed by Francis Galton and Karl Pearson, were conceptually better suited for the analysis of observational studies rather than experiments. Indeed, these tools were developed for surveys of demographic data, or mapping of human characteristics (such as height or weight) in very large populations.

In contrast, biological experiments presented with themselves a different set of challenges. The essence of a statistical analysis of an experiment is to compare sets of observations (i.e., experimental groups versus control). In biological experiments, there quickly emerged a problem of the small numbers of individual observations in the sets compared. The numbers tended to be small for practical reasons, such as limited budgets, limited manpower, lack of space in experimental facilities, or difficulty in sourcing biological specimens. As such, small samples are associated with sampling errors (in addition to the measurement error and biological randomness). As a result, biometric methods, developed for large populations, were both too complicated and too inaccurate to be useful for biological experiments. The problem became known as the small sample problem. The problem was only properly addressed by the second generation of statistical tools, pioneered by William Sealy Gosset, and developed further by Ronald Fisher [13] (Fisher also initiated work on statistical tools for the analysis of experimental groups employing the analysis of variance—ANOVA—rather than more established analysis of means. Moreover, he begun work on a theoretical framework—Fisher information—for the prediction of minimal sample sizes necessary to detect experimental effects of a given magnitude.).

Note that many contemporary genomic applications work well within the Fisherian framework of small-sample-problem-aware parametric statistics. This is the case when one encounters typical measurement and sampling errors (for example, in microarray studies with several replicates in each experimental group).

In other situations, data analyzed are not a random sample from any population. Indeed, individual genome projects are not at all experiments. They are more similar to maps in cartography. In other words, sequences of individual genomes present themselves as they are—without any sampling error. That is to say, there is only one consensus genome sequence for the human species, and the goal of a genome project is to map the genome sequence completely and accurately. As there is no population of sequences that is sampled, the most fundamental assumptions of Fisherian parametric statistics are grossly violated.

Fortunately, genomics datasets emerged at the end of the 20th century, when substantial computational resources were easily available—even on office computers. Presently, even laptop computers are frequently sufficient to analyze genomics datasets. This makes it rather practical to apply permutation and randomization methods, which are free from the assumption of random sampling. Such methods were known since Fisher’s times, but were initially rarely used due to their computational intensity [20]. In the case of observational studies, one can apply a permutation procedure to compute the distribution of the test statistic under the null hypothesis. Typically, a subset of genomic features is compared against the background of the features in the whole genome (In the case of experimental studies, one can employ a randomization test [21], in which observations are randomly assigned to experimental groups, or the control.).

The problem of multiple testing was another statistical challenge frequently met in functional genomics. For example, there are as many tests as microarray features when comparing sets of microarrays to identify differentially expressed genes. The problem was quickly recognized in the early days of functional genomics at the beginning of the 21^st^ century. As a response, robust and well-characterized solutions for the problem of multiple testing emerged, such as procedures that control the rate of false discovery [20].

The current trend in genomics is to transform it to a data science. Increasingly, the most recent computer science algorithms, such as deep learning, are applied to genomics datasets. We must however be aware that data science emphasizes practical algorithms, but neglects statistical theory and statistical inference. On a positive side, we can enjoy pragmatic benefits delivered by powerful industrial-grade algorithms. On a negative side, there is relatively little understanding of how exactly these algorithms work, what is their sensitivity and specificity, or against what kinds of inputs they are likely to fail.

### 1.11. Scientific Model Building

A scientist is frequently attempting to construct a model for an unknown mechanism [2], for example to interpret observations or experiments. Scientific modeling is a creative process—an art for which there are no strict guidelines or firmly established rules. Note that models are never perfect representations of the material world. They contain many assumptions, idealizations, and simplifications. Nonetheless, models can be very useful in science if used skillfully for the right purpose.

A model is a physical, conceptual, mathematical, or probabilistic approximation of a real natural phenomenon that is of interest to a scientist. In general, the advantage of constructing a model in science is that it is easier to work with than the mechanism being modeled. One can understand, visualize, study, or manipulate a model easier than the raw mechanism. One can also use a model to make abstract mechanisms more concrete. Moreover, a model can be used to discover, that is to get to know further mechanistic details of the phenomenon under investigation. Finally, a model can be used to predict a future behavior of the system.

I will now give an example of a very simple but, nonetheless, extremely influential model in physics. This is the Copernican model of the solar system. It is frequently said that Copernicus completely changed the paradigm of astronomy by putting the sun in the center of the solar system [15,22]. This changed the long established orthodoxy of the geocentric model of the Universe due to an ancient astronomer and mathematician—Claudius Ptolemy (The Ptolemaic model had Earth at the center.). However, it is less known that the model of Copernicus was so simple that it did not actually produce better predictions for astronomical data. At the same time, a major advantage of the Copernican system lied precisely in its geometrical frugality (That is to say, the Copernican model explained observed data in more parsimonious terms.).

Intellectually, the achievement of Copernicus lied in having the courage to question a long established orthodoxy that was favored by dominant non-scientific ideologies of the time. Only later, astronomers proficient at empirical observations and mathematics, such as Galileo, Kepler, and Newton, provided solid results in support of the heliocentric model. Moreover, additional details of the model were provided. Circular orbits were replaced with elliptical ones, and the force of gravity was proposed to explain the movements of planets.

Unfortunately, scientific modeling is less established in biology than in astronomy and physics. Moreover, the term *model* is loosely defined in biology. There is also an overlap between how the terms *model* and *concept* are used. A key to being a successful biologist may lie in choosing an appropriate modeling technique for the mechanism of interest, and for the purposes to which the model is to be applied. For example, when applied to the gene concept, modeling techniques differ in the level of mechanistic detail included (see Table 1). At one end of the spectrum, the models could be like the Mendelian black box—where nothing is known about the mechanism hidden inside. Alternatively, the models could be like rough sketches characteristic of classical genetics. Finally, there are also detailed see-through models characteristic of molecular biology (such models resemble transparent glass boxes).

Nonetheless, models can be very useful in biology as well. For example, Watson and Crick famously used modeling to discover the structure of DNA. At first, Watson and Crick constructed two-dimensional models of individual DNA bases. They manipulated the bases manually—seeking to understand if and how they could pair. Later, they also constructed a three-dimensional (3D) model of the double helix to visualize the pairing of the strands of DNA (Admittedly, it is true that the model of Watson and Crick was critically informed by Rosalind Franklin’s original data on X-ray diffraction patterns generated by purified DNA [24]. Nevertheless, Watson and Crick did show more intellectual courage and good judgment when constructing their comprehensive 3D model of DNA.). This, in turn, suggested a likely mechanism of DNA replication. This is just one example of how important modeling can be as a part of logical analysis, providing generalizations and concepts as added value to raw empirical observations.

## 2. Practical Examples of Genomic Analyses

A few published examples will be given and discussed. The examples will be re-interpreted in the light of the theoretical material discussed in Part I. In particular, I will ask whether a genomic analysis under consideration followed reductionist or anti-reductionist principles. I will also inquire whether the empirical method employed was an observational survey—conceptually analogous to a map. Alternatively, an analysis might have been more akin an experiment—conceptually analogous to a test of hypothesis (A summary of the examples is given in Table 2.).

### 2.1. A Survey of Individual Endothelial-Specific Genes (Which Followed Reductionist Principles)

Let me start with some background information necessary to understand this example. Vertebrates, a clade that includes the human species, are animals with complex body plans and hundreds of different cell-types. Although, evolutionary biologists do not like to designate any taxon as more advanced than others, it is a fact that vertebrate animals have more tissue- and cell-types than any other group of organisms on Earth. How do vertebrate cell types differ between each other? The answer is that all cells in the human body have the same genome, however, different cell types follow different differentiation trajectories. During differentiation, the epigenome is modified and different sets of genes are sequentially activated to be transcriptionally expressed. Thus, differential expression of genes defines individual cell types. This is true both during development and in terminally differentiated somatic cells.

The example under consideration in this section focuses on one of somatic cell types. This is the endothelial cell (EC) type, which is spread throughout the body, being present in all tissue types. The endothelium is a single layer of cells lining the lumen of the cavities of blood vessels. Note that evolutionarily ECs are unique to vertebrates, as there is no true endothelium in invertebrates [32] (Remarkably, the endothelium is present in every vertebrate species without exception. This is because the endothelium emerged along with the pressurized circulatory system characteristic of the vertebrate clade.).

The endothelium plays a primary structural role in maintaining the integrity of blood vessels. At the same time, ECs are not just a passive structural barrier. These cells have functional roles in addition to their structural role within the vasculature. For example, ECs are a primary instrument of angiogenesis—that is the process of the sprouting and growth of new blood vessels from pre-existing blood vessels. However, the EC cell type is remarkably active, taking part in many physiological processes besides angiogenesis. These processes include the regulation of vascular permeability, the control of hemostasis, the regulation of blood pressure, as well as the recruitment of immune cells. The versatility of ECs is reflected in a rich set of endothelial transcripts, many of which are preferentially expressed in ECs or even entirely specific to this cell type.

Inspired by the fact that the rich repertoire of genes expressed in ECs characterizes this cell type, Huminiecki and Bicknell set out to identify through transcriptomics the most endothelial-specific genes. Broadly speaking, their strategy was an integrative meta-analysis of functional genomic databases followed by experimental verification [27]. In technical terms, the analysis consisted of two parts. The first part was a computational meta-analysis of pooled datasets generated from a number of libraries based on several different genomic technologies. The datasets were available in the public domain (Note that the datasets analysed in the computational part included genomic surveys, as well as genomic experiments. For example, one of the experiments in the meta-analysis was a comparison of transcriptomes from the cell culture of human microvascular ECs with or without angiogenic stimulation. The angiogenic stimulation consisted of cell culture in the medium with added vascular endothelial growth factor. In the surveys, only libraries from ECs cultured in standard conditions were analysed: there were no additional experimental factors.). The goal of the meta-analysis was to computationally identify consensus endothelial-specific genes. The consensus predictions were then prioritized for empirical verification. In the empirical verification, RNAs from a small panel of endothelial and non-EC cell types were used to test whether the consensus predictions were indeed endothelial-specific in their spatial expression pattern.

Conceptually, I argue that Huminiecki and Bicknell took a reductionist approach to the analysis of the endothelial transcriptome. This is because the authors strived to identify individual genes that were expressed most specifically in this cell type. Crucially for the reductionist argument, the authors assumed that the essence of the endothelial transcriptome could be discerned by looking at individual genes independently of each other—rather than by looking at signaling or metabolic pathways or networks. Moreover, Huminiecki and Bicknell assumed that sufficient insights could be derived just by looking at finally differentiated ECs, without an analysis of progenitor cells or differentiation trajectories (Note that an analysis of differentiation would be technically much more difficult. Such an analysis would have to include time courses and multiple cell types).

### 2.2. A New Genomic Technology for the Analysis of Individual Promoters

A reductionist wants to divide a problem into smaller and smaller parts. This strategy is frequently facilitated by the emergence of a new laboratory technology, which typically allows one to look at a given biological mechanism in more detail.

For example, surveying the mechanism of gene expression at a greater resolution recently changed scientists’ view on the diversity of the transcriptome. Namely, a new technology for the detection of transcription, called Cap Analysis of Gene Expression (CAGE), allowed scientists to map gene expression at the level of individual base pairs [33]. Previously, microarrays typically only measured the expression signal of whole genes, ignoring the fact that most genes have multiple alternative promoters and transcriptional start sites—TSSes. Thus, the essence of the technological advancement brought about by CAGE is that one can characterize individual TSSes across the entire genome. Therefore, there is more detail and no bias is introduced by pre-selecting genomic features.

In recent practice, CAGE technology was employed by an international research consortium for the functional annotation of the mammalian genome (FANTOM5), led from Japan, to map gene expression in human and mouse genomes [34]. The survey performed by FANTOM5 was comprehensive; consortium’s expression data included profiles of 952 human and 396 mouse tissues, primary cells, and cancer cell lines. The map of transcription generated by the consortium demonstrated that most mammalian genes have multiple TSSes (and accompanying promoters located in the brackets of 3000 base pairs upstream/downstream of the TSS [26]). Moreover, alternative promoters can differ in expression patterns they drive [34], and in the structure of resulting transcripts (Interestingly, some of the variability in transcripts produced was previously known in the literature. But the variability was attributed to alternative splicing rather than to the existence of alternative TSSes.).

Furthermore, the map of gene expression generated by FANTOM5 became a reductionist research tool for dozens of other projects. For example, researchers used this dataset to map a correlation between the size of the architecture of a promoter, and the breadth of expression pattern the promoter drives [25]. In a further reductionist step, it was even technically feasible to survey separate contributions made to the housekeeping gene expression pattern by individual kinds of transcription factor binding sites located within the proximal promoter [26,35]. This, again, underlines the power of reductionist strategies in genomics. Indeed, many insights can be achieved by dividing the problem at hand into smaller and smaller parts, and analyzing it at a greater resolution.

### 2.3. An Anti-Reductionist Analysis of the Evolution of an Entire Signalling Pathway

In the next example, Huminiecki et al. [30] followed anti-reductionist principles in an evolutionary context. The approach could be dubbed: evolutionary systems genetics. The empirical method employed was a survey of genomes, conducted from an evolutionary perspective (rather than a test of an experimental hypothesis). Specifically, the paper focused on the emergence, development, and diversification of the transforming growth factor-*β* (TGF-*β*) signaling pathway.

Some background information is necessary to illuminate the narrative of the paper. It must first be mentioned that most genes belong to gene families, which are derived through consecutive cycles of gene duplication. In animals, gene duplication is the most important source of new genes and new cellular functions for the evolutionary process. However, genes usually duplicate individually: that is one at a time. Whole genome duplication (WGD) is a rare and dramatic evolutionary event, a so-called macro-mutation. In a WGD, all genes duplicate simultaneously (this is known as polyploidization). In 1970, Susumu Ohno suggested that a WGD event, termed 2R-WGD, occurred at the base of the evolutionary tree of vertebrates [36]. More recently, the 2R-WGD hypothesis received a lot of empirical support from genome sequences analyzed using sophisticated algorithms aiming to detect WGDs [37].

Huminiecki et al. also investigated the 2R-WGD hypothesis using genome sequences, but the authors focused exclusively on the components of the TGF-*β* pathway. Altogether, there are eight intra-cellular transducers (Smads) for the pathway in the human genome, accompanied by twelve different TGF-*β* receptors. After a survey of homologs of these human genes in 33 animal genomes, Huminiecki et al. deduced using the principles of parsimony that the evolutionary emergence of the TGF-*β* pathway paralleled the emergence of first animals. The pathway can be inferred to have initially existed in a simplified ancestral form, including just four trans-membrane receptors, and four Smads. This simple repertoire of the components of the ancestral pathway is similar to that observed in the extant genome of a primitive tablet animal in the species *Trichoplax adhaerens*. However, the pathway expanded in the evolutionary lineage leading to humans following 2R-WGD.

The interpretation of the above genomic screen focused on the following evolutionary hypothesis. The increase in the number of components of the pathway probably paralleled an increase in complexity of biochemical functions that the pathway could carry out, as well as an increase in cellular/organismal processes in which the pathway played a role. Note that progenitors of first animals were probably colony-forming organisms, similar to colonial choanoflagellates, with little specialization of cell-types. Accordingly, the primitive TGF-*β* pathway was probably only involved in sensing nutrients, or in mediating adhesion and attachment to solids (This ancestral function is probably still present in taxa Placozoa and Porifera.). However, the pathway gained an important role in cellular transfer of signals as numerous specialized cell types that emerged in true multicellular animals. (This function still corresponds to the role fulfilled by the pathway in invertebrates.) The TGF-*β* pathway further gained in complexity with the emergence of vertebrate animals—becoming a sort of super-signaling engine capable of communicating diverse stimuli and fulfilling a bewildering variety of biological roles. Indeed, the versatile vertebrate version of the pathway functions in many diverse physiological processes. These processes range from development, through organogenesis, to the control of stem cells, and even in immunity.

Note that the conclusions reached by Huminiecki et al. were only possible after the anti-reductionist analysis. The entire TGF-*β* pathway had to be analyzed in parallel—consisting of all its receptors and intra-cellular signal transducers in as many animal genomes as possible. A reductionist analysis of individual components, gene-by-gene or exon-by-exon, performed no matter with how much attention to detail, could not deliver the synthesis and relevant insights.

Another methodological point is that an evolutionary analysis of the type undertaken by Huminiecki et al. is not an experiment but instead a type of systematic and deliberate observational study, carried out on sequences from extant genomes. Needless to say, an experimental test of a hypothesis on such extensive evolutionary timescales would be impossible.

### 2.4. An Anti-Reductionist Analysis of the Evolution of the Entire Vertebrate Signalling Network

Following the aforementioned anti-reductionist analysis of the TGF-*β* pathway, Huminiecki and Heldin broadened their investigations. That is to say, the impact of 2R-WGD on the evolution of the entire vertebrate signaling network was analyzed in a follow-up paper [31]. Specifically, Huminiecki characterized in the follow-up paper the impact of 2R-WGD on the functionality of the whole cell (and the organism). What matters most for the purpose of the current review is that the above analysis had to be anti-reductionist. The global trends could have never been discovered by looking at individual genes in isolation.

The authors identified functional classes of genes that 2R-WGD had greatest impact on. In terms of their biochemical roles, these were signaling genes (i.e., ligands, receptors, intracellular transducers), as well as transcription factors effecting the responses of signaling pathways. In terms of the corresponding biological processes, genes preferentially retained following 2R-WGD provided emergent vertebrates with their specific evolutionary novelties. These included (1) the finely-tunable machinery of cell-cycle, (2) multi-compartment brains wired by neurons endowed with versatile synapses, (3) a pressurized circulatory system and a heart that powers it, (4) dynamic musculature and bones which facilitate active locomotion, and (5) adipose tissue that facilitates thermoregulation. Clearly, the above set of evolutionary novelties powered the radiation of vertebrates, and kick started their subsequent evolution on land.

Can the above trends be generalized to a universal scientific law? It turns out that the answer is ‘yes’, but only partially. Specifically, preferential retention of signaling genes and transcription factors after WGDs is a general law. This is because similar conclusions could be reached following analyses of WGDs in animals, plants, yeasts, and protozoans [37]. WGDs rather than duplications of individual genes facilitate the evolution of cellular network hubs and rewiring of the cellular network. At the same time, genomic evidence for WGDs can be only rarely observed in animals, but rather frequently in plants (This is probably due to reproductive differences.).

## 3. Conclusions

The take-home message is that the choice of an appropriate model of the gene strongly depends on the goals of a multidimensional analysis. This is because alternative models of the gene are on a broad spectrum strongly differing in the level of physical and chemical detail being modeled. The right approach will vary depending on many factors, for example depending on the sub-discipline of genetics, the hypothesis being tested, the design of the experiment, statistical methods used, and whether the study is of purely academic, translational, or industrial interest. Indeed, it is critical to choose the optimal level of abstraction for various genetical concepts in a computational project. The scientist and the statistician should discuss their strategy in advance. Such decisions are of equal importance to the choice of an optimal experimental design and a sensitive statistical test.

Indeed the alternatives available are broadly varied. The optimal choice of the model of the gene may fall somewhere on the spectrum close to radical reductionism (which is similar in spirit to the methods of molecular biology, biochemistry, or even organic chemistry). Such approaches will strive to directly model chemical and physical phenomena for individual molecules, or even atoms. At the extreme end of the spectrum, a computational geneticist might even strive to model the quantum phenomena within individual atoms (For example, if the intention was to understand how mutagens interact with the stacked arrangement of bases in nucleic acids).

Alternatively, it might be preferable to employ an anti-reductionist strategy. I illustrated this with examples from my own work, focusing on the TGF-*β* signaling pathway (example 3) or on the vertebrate signaling network (example 4). The examples served to illustrate the principle but were by no means comprehensive in their representation of the field. Indeed, various anti-reductionist strategies are currently widely employed in both biology and medicine. Analyses of different kinds of biological networks are being applied to either theoretical or practical problems. The networks include not only signaling pathways, but also protein interactions, metabolic pathways, or transcriptional networks. Indeed, there emerged whole new fields of research such as systems genetics, network biology [38], or network medicine [39,40]. For example, in systems genetics, databases of cellular protein interactions and signaling pathways are being used to model the inheritance of complex traits and to interpret the results of genome-wide association studies [41,42]. In cancer research, mutations are being put in network context to predict tumor subtypes, or to identify key signaling hubs that might be attractive targets for pharmacological anti-cancer interventions [43] (Note also that a cancer signaling map [44] was re-used by me in an evolutionary context in example 4.). Indeed, the anti-reductionist approaches vary widely over different inputs, across varying biological networks, and depending on accepted performance metrics. Therefore, benchmarking studies comparing the performance of different algorithms proved useful in highlighting their respective strengths and weaknesses [45,46].

Unfortunately, there is still not enough recognition in the literature of the theoretical importance of the choice of either a reductionist or an anti-reductionist agenda for data-mining. Moreover, there are few formal guidelines for the choice of the suitable model of the gene to fit the purposes of a given genomic analysis. This review hopes to be of some help in defining the challenge and setting up the stage for fuller theoretical and statistical considerations in the future.

## Figures and Tables

**Table 1 entropy-22-00942-t001:** Several modeling approaches have been applied to the gene concept.

Gene Concept	Modeling Approach	Empirical Evidence
Mendel’s cellular elements.	A black box model. The gene has certain functions, but nothing is known about the components of the genic mechanism, or how they interact.	Influenced by Darwinian natural history and associated 19th century evolutionary debates. Mendel performed experiments, but might selectively reported them [23].
The gene of classical genetics, and the chromosomal theory of heredity.	A sketch (i.e., a semi-transparent glass box). Mechanistic details are fuzzy, but the gene has several well-defined and empirically proven properties.	Initially, the evidence came from field experiments in plant genetics. Increasingly, there were also experiments in animal genetics for which specialized model systems were set up in laboratories (such as Morgan’s fruit fly model).
The gene of population genetics.	Mathematical equations. Statistical analyses. Probability theory.	The observations of genetic variability in natural or artificial populations. Some experiments, especially in the context of artificial evolution.
The molecular gene.	A transparent glass box.	Experiments and genetic engineering focusing on simplest organisms, to establish the basic principles. Next, the basic principles were extended to other species.
The gene of genomics.	A hierarchy of domains and functional units.	High-throughput screens from the surveys of populations, or from experimental groups.
The evolutionary gene.	The model includes information on gene’s evolutionary history, in particular on the pattern of duplications and speciation events. Data-mining strategies can be anti-reductionist, for example genes can be grouped and analyzed as gene families.	Morphological or sequence characters; gene and protein sequences can be aligned and phylogenetics trees can be constructed.
The virtual gene.	Computational data integration; data storage in relational databases. Data-mining strategies can be anti-reductionist, for example genes can be grouped and analyzed as pathways or networks.	Many kinds of empirical data can integrated using bioinformatics pipelines and databases. Data can be browsed using genome browsers, or data-mined using statistical tools and visualization.

**Table 2 entropy-22-00942-t002:** Examples of data-mining of genomics datasets. Examples of reductionist and anti-reductionist data-mining strategies in genomics are given. Analytical strategy and the focus of the analysis are specified, as well as the type of evidence and the main result.

Strategy	Focus. (Evidence)	Main Result	Reference
**Reductionist**	Promoter (An integrative survey of functional genomic datasets.)	A correlation between the size of promoter architecture and the breadth of expression was detected. Transcription factors with the strongest contribution to housekeeping expression were identified.	[25,26]
	Gene (A meta-analysis with experimental follow-up.)	First, an integrative data-mining procedure was used to clone the most endothelial-specific genes. The procedure was combined with experimental verification. Subsequently, one of the most endothelial-specific genes — ROBO4 — was found to be expressed in sites of active angiogenesis.	[27,28]
**Anti-reductionist**	Gene family (An integrative survey of multiple genomics databases.)	The analysis of the gene family of roundabouts suggested that magic roundabout (ROBO4) is an endothelial-specific ohnolog of ROBO1. ROBO4 neo-functionalized to an essential new role in angiogenesis	[29]
	Signaling pathway (A survey of 33 animal genomes.)	The evolution of the TGF-beta signaling pathway in the animal kingdom was analyzed. The components of the pathway were found to have emerged with the first animals, and diversified in vertebrates.	[30]
	Signaling network (An integrative analysis of an evolutionary database and a signaling network.)	2R-WGD was found to have remodeled the signaling network of vertebrates. This macro-mutation facilitated the evolution of key vertebrate evolutionary novelties and environmental adaptations.	[31]

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
