# Peer review of "Models of the Gene Must Inform Data-Mining Strategies in Genomics"

_entropy, 2020, doi:10.3390/e22090942_

Round 1

Reviewer 1 Report

The article gives an excellent overview of the history and thinking behind diverse approaches to the interpretation of genetic datasets. I think it is a valuable contribution to the conceptual understanding of biology in light of ever larger and more complex genomic data. The article encourages to careful consideration of the fact that specific mindsets and appropriate levels of abstraction for interpretation and analysis are necessary if one aims to understand specific biological phenomena. It is a well written, well structured article and I found it enlightening and entertaining to read and support publication without reservations.

Author Response

Dear Editor,

I would like to thank Referee 1 for a positive review and their recommendation to publish the manuscript.

I would also like to thank Referee 2 for their expert comments. Referee 2 makes a true point that relevant literature is much broader. This is correct. However, a thorough review of literature was never the goal of the manuscript. I rather wanted to make several theoretical points and to illustrate them with few selected examples. In particular, I strived to link current trends in computational biology to the long-standing philosophical debate on the gene concept. This is the goal of Part 1. In Part 2, examples were chosen mostly to illustrate the distinction between reductionist and anti-reductionist data-mining strategies.

However, I have significantly expanded the third paragraph in the Conclusions section to signal to the reader that there are currently a great many anti-reductionist approaches in computational biology and medicine. Perhaps this would be sufficient to put my effort in a broader context?

With best regards,

Lukasz

Reviewer 2 Report

The review is interesting, but I have some suggestions: 

One of the key concepts here is the systems biology and data integration. This part has not been well documented.  For example, molecular network has been used as platform for data integration and getting insights at the systems-level. More examples and concepts should be added. Related papers and reviews: PMID: 24075989; PMID: 22194470; PMID: 31701019; PMID: 24747696; PMID: 23791722; PMID: 23792107; PMID: 14735121; PMID: 31860671; PMID: 31765387; PMID: 32302302; PMID: 31479437; PMID: 31009458 and so on

Author Response

(The authors gave the same response as above.)
